# Clinical Practice of Photodynamic Therapy Using Talaporfin Sodium for Esophageal Cancer

**DOI:** 10.3390/jcm10132785

**Published:** 2021-06-24

**Authors:** Tomonori Yano, Tatsunori Minamide, Kenji Takashima, Keiichiro Nakajo, Tomohiro Kadota, Yusuke Yoda

**Affiliations:** Department of Gastroenterology and Endoscopy, National Cancer Center Hospital East, Kashiwa 277-8577, Japan; tminamid@east.ncc.go.jp (T.M.); ktakashi@east.ncc.go.jp (K.T.); knakajo@east.ncc.go.jp (K.N.); tkadota@east.ncc.go.jp (T.K.); yuyoda@east.ncc.go.jp (Y.Y.)

**Keywords:** photodynamic therapy, esophageal cancer, talaporfin sodium

## Abstract

Photodynamic therapy (PDT) using a conventional photosensitizer was approved for esophageal cancer in the early 1990s; however, it was replaced by other conventional treatment modalities in clinical practice because of the high frequency of cutaneous phototoxicity and esophageal stricture after the procedure. The second-generation photosensitizer, talaporfin sodium, which features more rapid clearance from the body, was developed to reduce skin phototoxicity, and talaporfin sodium can be excited at longer-wavelength lights comparing with a conventional photosensitizer. Endoscopic PDT using talaporfin sodium was initially developed for the curative treatment of central-type early lung cancer in Japan, and was approved in the early 2000s. After preclinical experiments, PDT using talaporfin sodium was investigated for patients with local failure after chemoradiotherapy, which was the most serious unmet need in the practice of esophageal cancer. According to the favorable results of a multi-institutional clinical trial, PDT using talaporfin sodium was approved as an endoscopic salvage treatment for patients with local failure after chemoradiotherapy for esophageal cancer. While PDT using talaporfin sodium is gradually spreading in clinical practice, further evaluation at the point of clinical benefit is necessary to determine the importance of PDT in the treatment of esophageal cancer.

## 1. Introduction

Photodynamic therapy (PDT) is a unique cancer treatment that involves a photosensitizer (PS), which is incorporated into cancer cells and activates light of a specific wavelength. The main mechanism of action of PDT for cancer tissue is known to be direct cytotoxicity due to reactive oxygen species, which occurs as a result of photochemical and photophysical reactions between PS and the specific light [1]. Esophageal cancer was, in the early 1990s, one of the first approved clinical indications of PDT using conventional porphyrin-based PS both in the United States and Japan [2]. In the United States, PDT has been approved as a palliative treatment for symptomatic obstructive advanced esophageal cancer, and eradicative ablation of high-grade dysplasia of the esophagus, which is a premalignant disease of esophageal adenocarcinoma. Contrarily, PDT was approved as a curative treatment for superficial esophageal cancer that was deemed difficult to remove with conventional endoscopic resection (ER) in Japan. However, PDT was replaced in clinical practice by other effective and conventional treatment modalities for esophageal cancer patients in both countries at the beginning of the 2000s. Radio-frequency ablation was established as the standard of care for the eradication of dysplastic Barret’s esophagus based on a large volume of evidence, including a randomized control trial [3]. The use of endoscopic submucosal dissection (ESD), an evolved technique of ER that can remove large superficial esophageal cancer, has dramatically increased worldwide [4]. The most significant reasons that PDT using porfimer sodium lost the opportunity to become a treatment for esophageal cancer patients were phototoxicity due to the PS and esophageal stricture after PDT. Therefore, the development of a PS that clears more rapidly from the body, targeted at clinical applications so that PDT can take advantage of its strength, was necessary in order for this to become a clinical practice for esophageal cancer patients. In this article, we review the history of PDT using talaporfin sodium, a second-generation photosensitizer, and its progression into clinical practice for use in esophageal cancer cases.

## 2. The Characteristics of Talaporfin Sodium

As mentioned above, a disadvantage of PDT using conventional PS, including porfimer sodium, is phototoxicity, which requires more than one month of sunshade. The second-generation PS, talaporfin sodium (Laserphyrin for Injection, Meiji Seika Pharma; Meiji Seika Pharma, Tokyo, Japan), which features a more rapid clearance from the body, was developed to reduce skin phototoxicity in Japan. Talaporfin sodium was developed as a mono-L-aspartyl chlorin e6 (NPe6), a chlorin-based pure plant-origin agent (Figure 1) [5,6]. Talaporfin sodium can absorb light at longer wavelengths (664 nm) compared to light absorbed by conventional porfimer sodium (630 nm); therefore, it is expected to affect deeper tissues. In a comparative experimental study between PDT using NPe6 and hematoporphyrin derivative (HpD) in a mouse model, PDT using NPe6 demonstrated the disappearance of palpable tumors with extremely low normal skin photosensitivity compared to HpD [6]. The depth of tumor tissue injury was significantly increased in PDT using Npe6 in a cholangiocarcinoma cell line-mounted mouse model compared with PDT using HpD [7]. Furthermore, considering the unique mechanism of blood flow stasis due to a significant reduction in the diameter of the lumen of arterioles, as well as direct cytotoxicity in rat models, the best tumor response was produced when the maximum effect of both vascular stasis and direct cytotoxicity was achieved in PDT using NPe6 [8]. A phase-I trial of PDT for patients with a variety of cutaneous cancerous lesions was investigated to assess the optimal dose of NPe6, as well as the safety and efficacy of PDT [9]. Twelve patients who refused or failed conventional treatment for metastatic or primary superficial cutaneous carcinoma, including head and neck squamous cell carcinoma or ductal adenocarcinoma of the breast, were enrolled in the study. A dose of 0.5–3.5 mg/kg of NPe6 was administered 4 h before laser illumination, and an argon-ion laser was used to activate PS at 664 nm. Flat tumors less than 1.5 cm in thickness were illuminated superficially using a microlens frontal fiber, and tumors with a thickness between 1.5 and 2.0 cm were illuminated interstitially using a cylindrical diffuser, while tumors more than 2.0 cm thick were not indicated. In this study, skin photosensitivity and phototoxicity were precisely assessed using a solar simulator. Phototoxicity was scored at frequent intervals within one week, and then every week until six weeks after drug administration. The NPe6 dose of 2.5 to 3.5 mg/kg combined with 100 J/cm^2^ light resulted in a 66% complete response (CR) and patients remained tumor-free for 12 weeks after treatment. Furthermore, apparent selectivity for destruction or necrosis was not shown between tumors and normal skin at the aforementioned doses of NPe6. PDT using NPe6 demonstrated a reduced degree and duration of cutaneous phototoxicity compared with conventional PS, even at a dose of 3.5 mg/kg, while patients treated with 3.5 mg/kg NPe6 showed only minimal erythema within 96 h after injection [9]. This phase-I trial demonstrated the potential of NPe6 as an effective and less cutaneous phototoxic PS in cancer patients.

In Japan, Dr. Kato and Dr. Furukawa, along with their team members, developed a strategy for endoscopic application of PDT using talaporfin sodium as a curative treatment for central-type superficial lung cancer. They introduced a diode laser of 664 nm as a light source and performed a phase-I trial for patients with bronchogenic early squamous cell carcinoma [10]. An excellent CR rate (87.5%, 7/8) was achieved without serious adverse events in the phase-I trial; therefore, they conducted a multicenter phase-II trial to evaluate the safety and efficacy of PDT for patients with lung cancer [11]. Patients who were not candidates for surgery or refused surgery for lesions of 2 cm or less in size with histologically confirmed squamous cell carcinoma were eligible. They set the treatment dose of talaporfin sodium as 40 mg/m^2^ and laser illumination at a power density of 150 mW/cm^2^ and energy level of 100 J/cm^2^ at 4 h after administration, according to the results of their phase-I trial. Forty-one patients with 46 lesions were enrolled in the study, and the CR rate was 84.6% of the evaluable lesions. Skin photosensitivity disappeared within two weeks of drug administration in 84.8% of patients. Based on the excellent results of this phase-II study, PDT using talaporfin sodium and a diode laser was approved for lung cancer in Japan in 2004. The endoscopic application of PDT using talaporfin sodium is expected to expand gastrointestinal cancer including esophageal cancer.

## 3. Innovation of PDT for Local Failure after Radiotherapy for Esophageal Cancer

Among the esophageal cancers treated with PDT using porfimer sodium in the past, local failure was the first target for the development of PDT using talaporfin sodium. Chemoradiotherapy (CRT) has demonstrated favorable results in patients with locally advanced-stage esophageal cancer or in those who declined to be treated with an esophagectomy at the operable stage. However, more than half of the patients in this study developed local failure at the primary site of the esophagus, and the survival outcome of these patients was extremely poor [12]. Therefore, the improvement of local control was an unmet issue in the non-surgical intervention for patients with esophageal cancer. We introduced PDT using porfimer sodium (Photofrin, Pfizer Japan Inc., Tokyo, Japan) and a 633 nm excimer dye laser (EDL-1; Hamamatsu Photonics, Hamamatsu, Japan) as a treatment option for patients with local failure within the T2 stage without any metastasis, and we carried out a single-center phase-II study to evaluate the efficacy and safety of PDT [13]. PDT was performed with the intravenous administration of 2 mg/kg of porfimer sodium 48–72 h after excimer dye laser illumination with a fluence of 75 J/cm^2^. Twenty-five patients were enrolled in the study, and 19 patients achieved CR resulting in a CR rate of 76% (95% CI; 55–91%). All patients were instructed to avoid sunlight or strong artificial light for one month after porfimer sodium administration; however, cutaneous phototoxicity was observed in eight patients (32%). We experienced a case of treatment-related death due to gastrointestinal hemorrhage at the treated site 33 days after PDT. We also reported favorable long-term survival outcomes of patients treated with PDT using porfimer sodium for local failure after CRT for esophageal squamous cell carcinoma in a retrospective study at a single institution [14].

PDT using talaporfin sodium as a salvage treatment for local failure after CRT for esophageal cancer.

To resolve the cutaneous phototoxicity of PDT using porfimer sodium, we would like to introduce talaporfin sodium to patients with esophageal cancer. The excimer dye laser, which excites the porfimer sodium, was extremely large and expensive, and its release was canceled due to the lower popularity of PDT using porfimer sodium in the early 2000s in Japan. A diode laser with a wavelength of 664 nm, which can excite talaporfin sodium, was portable and cheaper than the conventional system (Figure 2). Therefore, we attempted to introduce PDT using talaporfin sodium for esophageal cancer in a preclinical study. Tissue damage to the esophagus due to PDT using talaporfin sodium followed by 640 nm diode laser illumination through gastrointestinal endoscopy was evaluated in a living canine model, and tissue damage to the normal esophagus was deeper and larger as the illuminated amount of laser increased [15]. Preclinical study results suggested it would be desirable to investigate the optimal laser dose by initiating the minimal dose to increase step-by-step in human application. As such, a phase-I study to determine the optimum laser fluence rate commenced [16]. This laser dose-escalation study used a 40 mg/m^2^ fixed dose of talaporfin sodium, and the starting fluence of the 664 nm diode laser was 50 J/cm^2^, with an escalation plan to 75 J/cm^2^ and 100 J/cm^2^ with three patients at each level. In this phase-I trial, there were no cases of dose-limiting toxicity at any level of the laser dose. Furthermore, five of the nine patients with local failure after CRT for esophageal cancer achieved CR after PDT [16]. A multi-institutional phase II study was conducted to evaluate the efficacy and safety of PDT using talaporfin sodium followed by diode laser illumination for patients with local failure after radiotherapy or CRT for esophageal cancer [17]. Patients with histologically confirmed residual or recurrent esophageal squamous cell carcinoma (ESCC) at the primary site, limited to the T2 stage by endoscopic ultrasound, were eligible. Twenty-six patients with 28 histologically confirmed lesions (T1b: 21 and T2: 7) were enrolled, and all were treated with PDT. Of these patients, 23 with 25 lesions achieved CR (CR rate: 88.5%). There were no cases of cutaneous skin toxicity even with a two-week sunshade period and no cases of severe adverse events, including treatment-related death [17]. According to the results of this trial, PDT using talaporfin sodium and a diode laser was approved as an endoscopic salvage treatment for patients with esophageal cancer with local failure after radiotherapy or CRT in Japan in 2015.

## 4. Indication of PDT for Local Failure after Radiotherapy for Esophageal Cancer

The approved indication of PDT using talaporfin sodium for esophageal cancer is limited to patients with local failure after radiotherapy or CRT. A clinical flowchart for the indication of PDT using talaporfin sodium is presented in Figure 3. First, lymph node or distant metastasis should be evaluated using computed tomography; PDT is not indicated if the patients have metastases. The contraindication of PDT using talaporfin sodium was a history of hypersensitivity to talaporfin sodium or concomitant porphyria. Because of our experience of treatment-related death with gastrointestinal hemorrhage due to aortic rupture in patients treated with PDT using porfimer sodium [13,14], patients whose lesions before CRT were judged to involve the aorta were set as contraindications for PDT using talaporfin sodium. When patients do not have metastasis, the indication for ER, including ESD, should be evaluated because ER is the least invasive treatment for local failure lesions [18,19]. ER, including ESD, is generally indicated for lesions within T1a in depth without ulceration or severe fibrosis; therefore, the indication of ER is limited and must be judged based on the operator’s skill. While salvage surgery is the most invasive treatment method for patients with local failure after CRT for esophageal cancer, it can be a curative intent treatment when complete resection is achieved without any fetal complications [20,21]. Therefore, a patient’s physical tolerability for surgery should be discussed with surgeons at the conference, presenting the advantages and disadvantages of salvage surgery for patients. After these steps, if the patient is judged to be intolerant to surgery or the patient refuses surgery, PDT using talaporfin sodium can be an alternative treatment option. The definitive indication criteria of PDT in practice follow the eligibility criteria of a phase-II study [16], as follows: (1) local failure lesion that is suspected to be limited to within the muscularis propria (T2) depth of the esophageal wall; (2) longitudinal lesion length of 3 cm or less; (3) half circumference of the lumen or less; and (4) no invasion of the cervical esophagus. The depth of invasion was carefully evaluated using endoscopic ultrasonography and computed tomography. The limitation of the size of the criteria was determined to prevent esophageal stricture, which is the most significant major complication of PDT using porfimer sodium for esophageal cancer in previous reports [22,23]. Because of the anatomical stricture at the cervical esophagus or gag reflux, it is difficult to keep the per-oral endoscopy in the appropriate position that enables adequate laser illumination to the target lesions. PDT is not advocated if the lesions have invaded the cervical esophagus.

## 5. Clinical Practice of PDT Using Talaporfin Sodium for Esophageal Cancer

In PDT using talaporfin sodium for esophageal cancer, we can only use the frontal diffuser; therefore, delivering an adequate dose of laser illumination while maintaining the direct light viewing angle of endoscopy and the appropriate distance between the tip of the diffuser and the lesion is important. At the appropriate angle and distance, a plastic hood was attached to the tip of the endoscope, and the edge of the hood was fixed to the surface of the esophageal mucosa during illumination. Target illumination and a reduction in the influence of normal mucosa should be specifically focused on in order to avoid esophageal stricture after PDT. The illumination was initiated at the distal end of the lesion; the spot was moved to the oral side of the lesion step-by-step, and the fluence of the diode laser was set at 100 J/cm^2^ with a fluence rate of 150 mW/cm^2^. If the lesion was larger than 1 cm^2^, multiple treatment fields were overlapped to cover the entire lesion. An opaque hood is attached at the tip of the endoscope to shield the surrounding normal mucosa from the illuminated laser in order to prevent esophageal stricture (Figure 4).

After approval for clinical usage in Japan, PDT using talaporfin sodium followed by diode laser illumination for local failure after CRT or radiotherapy for esophageal cancer is gradually spreading, and there are several reports of the single-center experiences [24,25,26,27]. Minamide et al. reported a retrospective comparative study between PDT using talaporfin sodium and porfimer sodium in patients with local failure after CRT [26]. The local complete response was better in patients treated with PDT using talaporfin sodium compared with PDT using porfimer sodium (69.0% vs. 58.1%), although the difference was not statistically significant. Regarding the safety of PDT, both skin phototoxicity (4.5% vs. 18.2%, *p* < 0.049) and esophageal stricture (4.5% vs. 36.4%, *p* < 0.001) after PDT were significantly lower in patients treated with talaporfin sodium compared with those treated with porfimer sodium. PDT using talaporfin sodium was introduced to clinical practice later than that of porfimer sodium; therefore, the improvement of patients’ selection or technical tips could overestimate the clinical results of PDT. However, the study demonstrated a more favorable outcome following treatment with PDT using talaporfin sodium compared with conventional PDT. Amanuma et al. reported that a local complete response with PDT using talaporfin sodium was associated with good prognoses in patients with local failure after CRT for esophageal squamous cell carcinoma [27]. The local complete response was reported as 68% (23/34) in all patients (81% for T1 lesions and 46% for T2 lesions), and the median progression-free survival was 21.2 months in patients who achieved CR with PDT and 1.9 months in patients who did not achieve CR. Furthermore, the overall survival at two years was significantly better in patients who achieved CR with PDT than in patients who did not achieve CR (79%, 95% CI: 54–92% vs. 40%, 95% CI: 11–68%, *p* = 0.0389). These results highlight the importance of good local control in clinical oncology for esophageal cancer, and that PDT using talaporfin sodium could contribute to the survival benefit of patients with esophageal cancer.

## 6. Future Perspective and Conclusions

PDT could allow for the opportunity to treat esophageal cancer patients through the introduction of more rapid clearance of talaporfin sodium and the determination of appropriate targets for PDT. With an aging society, PDT will play a more valuable role in the non-surgical treatment of patients with local failure after radiotherapy for esophageal cancer. In order to encourage the clinical application of PDT in practice, several developments or clarifications in the practice of gastrointestinal oncology are necessary. First, the long-term survival benefit or treatment efficacy of PDT using talaporfin sodium, and the efficacy and safety of multiple sessions of PDT, have not yet been clarified. The long-term outcomes of the enrolled patients in phase-II trials will open soon, and other studies of sequential PDT for local failure after initial PDT are warranted in the future. Next, PDT is known to cause acute inflammation, leading to immune stimulation and increasing the presentation of tumor-derived antigens to T-cells [28]. PDT using third-generation PS of glucose-conjugated chlorin demonstrated a stronger antitumor effect and induced immunogenic cell death compared with talaporfin sodium in a preclinical study [29]. Several immune checkpoint inhibitors have shown more favorable survival outcomes for esophageal cancer patients compared with conventional systemic chemotherapy in phase-III trials, and these have been introduced into the clinical practice for advanced esophageal cancer patients in recent years [30,31]. The synergistic effect of PDT and immune checkpoint inhibitors has been investigated in preclinical studies for several cancers [32]. The combination of PDT and immune checkpoint inhibitors is an alternative treatment that should be evaluated in clinical practice for esophageal cancer patients. Third, there are several reports on the clinical application of PDT using talaporfin sodium for gastrointestinal cancer out of esophageal cancer [33]. A case series of PDT using talaporfin sodium for patients with gastric cancer or cholangiocarcinoma has been reported [34,35], and these studies represented the favorable outcome of PDT for patients. Further exploration is necessary to determine the appropriate target for PDT in clinical practice for these types of gastrointestinal cancers.

In conclusion, PDT using talaporfin sodium showed favorable outcomes for esophageal cancer patients; further evaluation of PDT for esophageal cancer at the point of clinical benefit is necessary to determine the importance of PDT in the treatment of esophageal cancer.

## Figures and Tables

**Figure 1 jcm-10-02785-f001:**
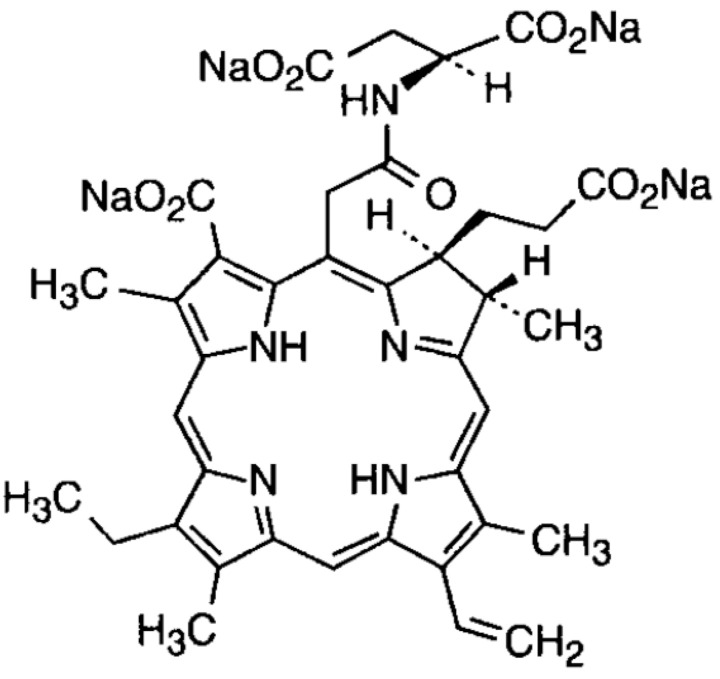
Structural formula of talaporfin sodium.

**Figure 2 jcm-10-02785-f002:**
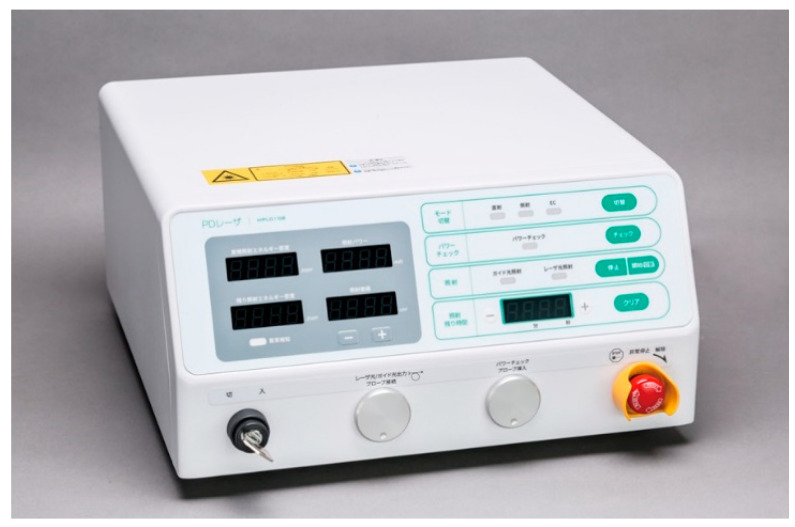
The system of a diode laser which can excite talaporfin sodium.

**Figure 3 jcm-10-02785-f003:**
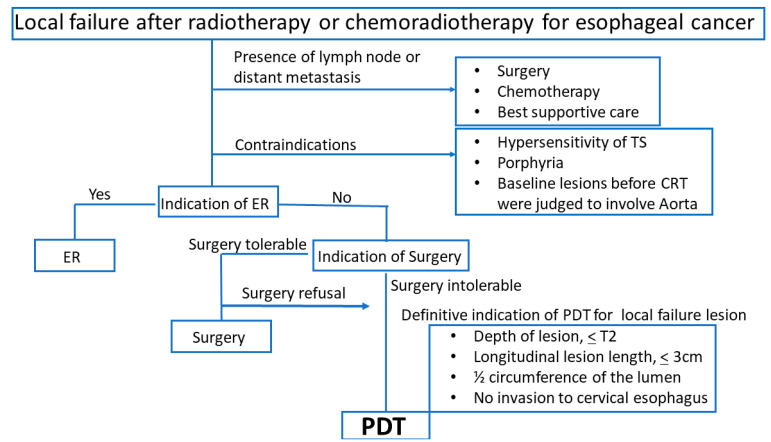
A clinical flowchart for the indication of PDT using talaporfin sodium for esophageal cancer.

**Figure 4 jcm-10-02785-f004:**
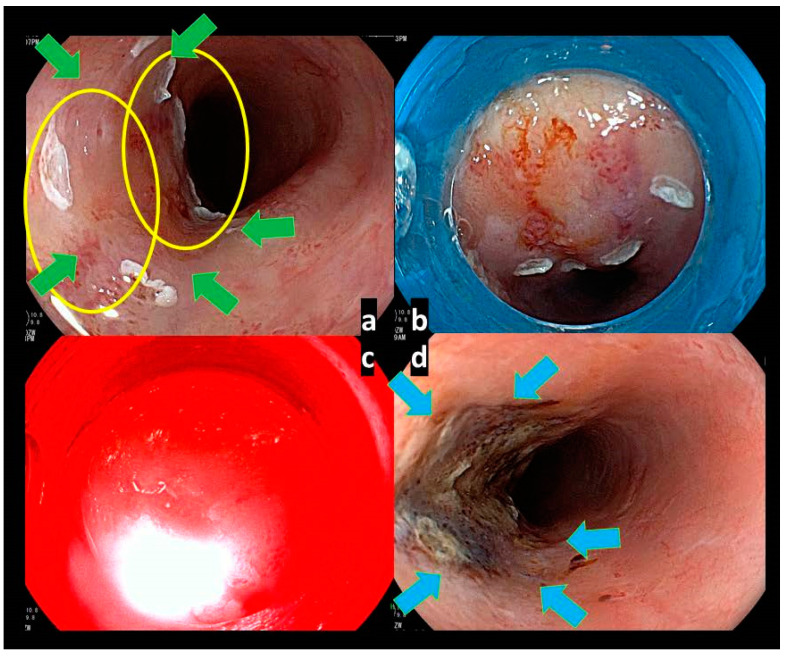
Endoscopic images of PDT procedure for local failure lesion after chemoradiotherapy for esophageal cancer. (**a**) The local failure lesion; we planned to illuminate laser the for two spots (yellow circles) to cover whole the lesion (identified with green arrows). (**b**) An opaque hood was attached at the tip of the endoscope to shield the surrounding mucosa during the illumination in order to avoid esophageal stricture. (**c**) The fluence of the diode laser at 100 J/cm^2^ with a fluence rate of 150 mW/cm^2^ was delivered through endoscopy, keeping a distance of less than 2 cm from the surface of the mucosa. (**d**) Endoscopic image a day after PDT showing the ischemic change at the laser-illuminated mucosa (identified with blue arrows).

## Data Availability

The study did not report any data.

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
