# Peer review of "Clinical Practice of Photodynamic Therapy Using Talaporfin Sodium for Esophageal Cancer"

_jcm, 2021, doi:10.3390/jcm10132785_

Round 1

Reviewer 1 Report

The authors reviewed the photodynamic therapy for esophageal cancer, discussing its clinical impact and history.

This review was interesting and informative, feel worth publishing. I have minor points for further improvement of this manuscript for publication.

Minor comments:

The term "conventional PS" has been used several times. I think it would be easier to understand if you specify what exactly you mean by conventional. Did the authors want to refer to porphyrins?

About the following subheadings;

Innovation of PDT using talaporfin sodium from the experience of PDT using conventional PS for local failure after radiotherapy for esophageal cancer

The detailed history is easy to understand.

I would like to recommend starting with a very short phrase describing what was the most innovative for better understanding.

The authors described that

“The optimum laser dose was determined at 100 J/cm2, and as promising anti-tumor effects were observed in the phase I trial, a multi-institutional phase II study was conducted to evaluate the efficacy and safety of PDT using talaporfin sodium followed by diode laser illumination for patients with local failure after radiotherapy or CRT for esophageal cancer [16].”

This sentence seems long. Please consider dividing this sentence into two sentences.

The authors described that

“First, lymph node or distant metastasis with should be evaluated using computed tomography …”

Is it possible to omit "with"?

I think it would be more grammatically correct.

P. 5/10 L. 173-174

Regarding the term "baseline” lesions before CRT was a bit difficult to understand.

I am wondering if the authors could rephrase the expression "baseline"

Author Response

The authors reviewed the photodynamic therapy for esophageal cancer, discussing its clinical impact and history.

This review was interesting and informative, feel worth publishing. I have minor points for further improvement of this manuscript for publication.

Minor comments:

The term "conventional PS" has been used several times. I think it would be easier to understand if you specify what exactly you mean by conventional. Did the authors want to refer to porphyrins?

In this article, the term “conventional PS” means porphyrins based photosensitizer “porfimer sodium”, therefore, I revised the term to be easier to understand.

About the following subheadings;

Innovation of PDT using talaporfin sodium from the experience of PDT using conventional PS for local failure after radiotherapy for esophageal cancer

The detailed history is easy to understand.

I would like to recommend starting with a very short phrase describing what was the most innovative for better understanding.

I appreciated for the recommendation of the subtitle, I would like to revise the subtitle to “Innovation of PDT for local failure after radiotherapy for esophageal cancer”.

The authors described that

“The optimum laser dose was determined at 100 J/cm2, and as promising anti-tumor effects were observed in the phase I trial, a multi-institutional phase II study was conducted to evaluate the efficacy and safety of PDT using talaporfin sodium followed by diode laser illumination for patients with local failure after radiotherapy or CRT for esophageal cancer [16].”

This sentence seems long. Please consider dividing this sentence into two sentences.

 Thank you for your advice, I deleted the former part of this sentence.

The authors described that

“First, lymph node or distant metastasis with should be evaluated using computed tomography …”

Is it possible to omit "with"?

I think it would be more grammatically correct.

I deleted the “with” in this sentence.

  1. 5/10 L. 173-174

Regarding the term "baseline” lesions before CRT was a bit difficult to understand.

I am wondering if the authors could rephrase the expression "baseline"

I deleted the “baseline” in this sentence.

Reviewer 2 Report

A through and comprehensive review which is a little difficult to read due to massive informations within each paragraph. A subdivision of the text with more headlines and some pictures of the equipment would help. 

I need information on the illuminated size of the tissue in each firing, how the procedure is performed, the distance to the tumor tissue, how to shield the diffuser, toxicity, data on the producer.

The authors has cited permissions from authorities to use talaporfin without references - should be included

Author Response

A through and comprehensive review which is a little difficult to read due to massive informations within each paragraph. A subdivision of the text with more headlines and some pictures of the equipment would help.

I divided paragraph and subtitle and pictures of procedure.

I need information on the illuminated size of the tissue in each firing, how the procedure is performed, the distance to the tumor tissue, how to shield the diffuser, toxicity, data on the producer.

I added the figure of procedure with figure legend to explain how to perform the procedure.

The authors has cited permissions from authorities to use talaporfin without references - should be included

I added the references for the authorities to use talaporfin sodium in the revised version.